# Integrating national open databases for a comprehensive view on food systems, environment sustainability and health in Brazil

**Maria Julia Miele** [1,2]*, **Jacqueline Tereza da Silva**[3]*, **Renato Teixeira Souza**[2], **José Guilherme Cecatti**[2], **Barbara Teruel**[1]

**1** School of Agricultural Engineering, University of Campinas (UNICAMP), Campinas, São Paulo, Brazil, **2** Department of Obstetrics and Gynecology, University of Campinas (UNICAMP), School of Medical Sciences, Campinas, São Paulo, Brazil, **3** Division Agriculture and Food Systems, University of Edinburgh, Edinburgh, United Kingdom

* mmiele@unicamp.br (MJM); Jacqueline.silva@ed.ac.uk (JTdS)

## Abstract

This paper details the integration of open-source databases on food production and consumption, pesticide use, water and land use, and nutrient supply, segmented by year and region. The process of extracting, transforming, and loading information was divided into four phases: 1) water and land use, harvest, and nutrient metrics; 2) pesticide and crop records; 3) pesticide residues with legal limits and their environment risk; 4) food acquisition and consumption by region and year. This effort resulted in 48 years of agrifood system data from 114 datasets across eight public platforms, providing a comprehensive view of the variations in agricultural production and consumption in Brazil.

## Introduction

The United Nations publication, "Transforming our world: The 2030 agenda for sustainable development," calls for learning from the Millennium Development Goals and shifting our mindset to reduce our footprint and enhance sustainability through the 17 Sustainable Development Goals (SDGs) [1]. This work particularly aligns with four SDGs: Zero Hunger (Goal 2), Responsible Consumption (Goal 12), Climate Action (Goal 13), and Life on Land (Goal 15). These goals collectively focus on ensuring the sustainable provision of healthy food to secure the well-being of humanity and ecosystems.

The Food and Agriculture Organization of the United Nations (FAO) predicts that the world's population will exceed 9 billion by 2050 [2]. This growing population puts increasing pressure on agricultural system, leading to greater pesticide use to meet rising food demands. However, concerns have been raised about the potential risks associated with pesticide residues in food, posing challenges to both human and environmental health [3,4].

**Data availability statement:** All datasets utilized in this study are publicly accessible and have been comprehensively detailed in the Supporting information. The integrated dataset, Planetary Health: A Comprehensive View of Food in Brazil (PHFood Brazil), along with its codebook, is available from Mendeley at DOI: 10.17632/mt4mj23j73.1. The Supplementary Material includes direct hyperlinks to each of the 114 open-access datasets from eight public platforms, along with thorough descriptions of the data sources and access procedures.

**Funding:** Maria Julia Miele was supported by the São Paulo Research Foundation (FAPESP) under grant #2025/00722-3 (Fellowship - Associated Submission / FAS - Postdoctoral). This work was supported by the São Paulo Research Foundation (FAPESP) under grant #2020/09838-0 (BIOS - Brazilian Institute of Data Science).

**Competing interests:** The authors report there are no competing interests to declare.

The concept of "Planetary Health" integrates the health of humanity with the sustainability of Earth's ecosystems upon which it depends [5] Brazil, one of the largest food producers globally, has been one of the top consumers of pesticides since 2008 [6]. While pesticides play a significant role in combating hunger and poverty, their health and environmental impacts remain underexplored and inadequately addressed [7].

In light of the urgent need to address these challenges in food production, data science has emerged as a transformative tool for compiling and integrating information on nutrition and agricultural systems over time. This approach enables the tracking of food production trends, fostering a deeper understanding of past and present practices, and identifying opportunities for the future [8]. The definition of Big Data (BD) is intrinsically linked to processes such as data collection and processing, with its main characteristic being an integrated approach that unites various aspects of personalized information with a wide range of structured and unstructured data sources to provide continuous and long-term insights [9,10]. Additionally, the future of agriculture increasingly relying on large-scale research that supports data-driven decisions and predictive solutions – areas where agriculture is still in its early stages [11]. For example, one key application of BD is in maximizing the benefits of pesticides in crop protection, ensuring their efficient and safe use without harming the environment or human health [12].

Despite the wealth of data generated daily in various formats, the challenge lies in aggregating and integrating these datasets. This requires decision-making and data science skills, particularly in Extract, Transform, Load (ETL) processes, to build and load an agricultural BD repository [13].

This study aims to detail the methodological process of integrating databases on food production and consumption within the Brazilian context over time. Rather than presenting new empirical findings, the study focuses on the construction of a comprehensive data repository in Brazil, organized with national databases that host information on agrifood systems, sustainability, health, and climate change. The harmonization of data from multiple national sources enables the identification of important relationships between agricultural production, nutrient availability, and environmental sustainability, shedding light on the dynamics of food system changes in Brazil.

Ultimately, food system data is essential for implementing targeted solutions in the right places and assessing variations in the consumption of certain foods in response to changes in crop production.

## Materials and methods

This study provides a methodological foundation for the integration and harmonization of national open datasets related to food systems in Brazil by detailing the steps taken during data integration, reporting on the Extract, Transform, Load (ETL) process, which consists of extracting data from multiple databases, transforming it into harmonized formats, and loading it into a final integrated dataset. This process is aimed at enabling future analyses across time, regions, and thematic domains such as health, environment, and agriculture. The final product is titled *Planetary*

*Health: A Comprehensive View of Food in Brazil - PHFood Brazil*, will be integrated with additional information to monitor issues related to health, climate change, and agricultural production. It aims to support health and agriculture initiatives by employing innovative, state-of-the-art machine learning techniques. To facilitate the use of the final compiled dataset, a codebook was developed containing the description of each variable. Both the PHFood Brazil dataset and its codebook are published and available on an open platform.

To clarify understanding of the data linkage steps, we present the architectures describing the data ingestion and aggregation aspects developed over four phases (Fig 1). Throughout these phases, decisions regarding agricultural features, nutritional content, and health-related data were made collaboratively by the team and supported by technical literature.

For data mining, harmonization, analysis, and visualization, we used the R software version 4.3.3 (2024-02-29 ucrt) – "Angel Food Cake" (R Core Team, 2024)[14]. Several packages were utilized, including "readxl" and "readr" for importing

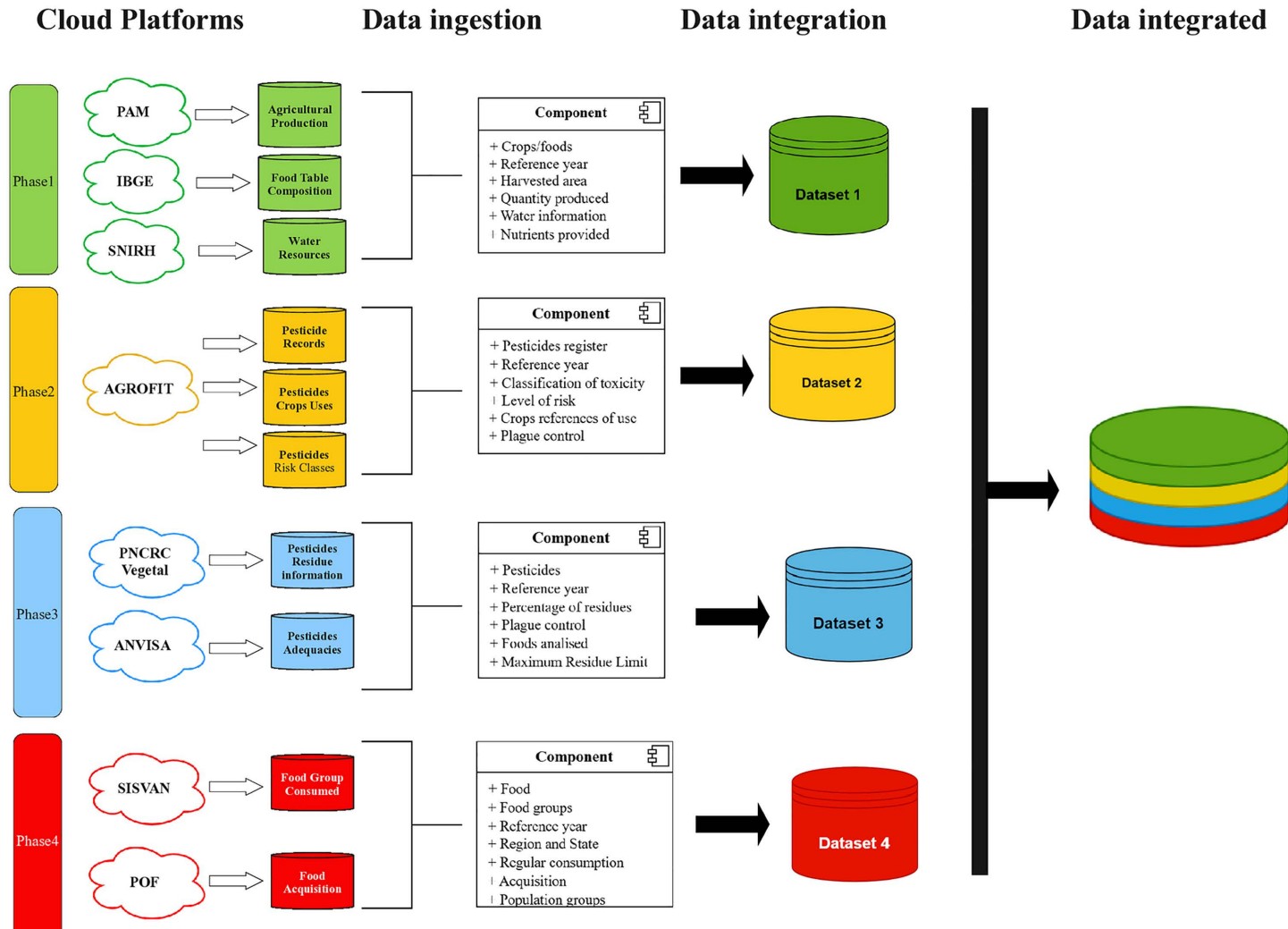

**Fig 1. Data Linkage and Aggregation Workflow.** This figure illustrates the sequential phases of data ingestion and aggregation, integrating information from multiple datasets to construct a comprehensive database.

data from Excel and CSV files; "dplyr," "magrittr," "tidyr," and "stringr" for data manipulation and transformation; "janitor" for data cleaning; "dplyr" and "fuzzyjoin" for merging datasets; and "writexl" for saving the datasets at each phase.

## Data compilation

**Agricultural data.** We compiled a harvest database using datasets from the Municipal Agricultural Production Information (PAM) platform, which provides detailed records of crop production in Brazil, including data on 33 permanent and 31 temporary crops from 1974 to 2022. The PAM platform primarily provides detailed information on crops harvested in Brazil, including 33 permanent and 31 temporary crops. Given the comprehensive nature of the PAM data and its focus on a wide range of crops, we decided to focus our analysis on vegetable crops. This decision was driven by the availability of consistent and detailed information for vegetables in the PAM datasets, which allowed for a more thorough and accurate analysis of agricultural production trends over time. More information is available at https://sidra.ibge.gov.br/pesquisa/pam/Tabelas.

Information related to agricultural pesticides, including year and classification, was taken from the Phytosanitary Pesticides System (Agrofit), and data on the percentage of residue adequacy was obtained from the National Plan for Control of Residues and Contaminants (PNCRC/Vegetal); both databases are available on the Ministry of Agriculture and Livestock (MAPA) platform. The Agrofit offers information on pesticides registered annually (1988–2023), categorized by functional classes, environmental impact, and toxicity per crop. More details from Agrofit: https://mapa-indicadores.agricultura.gov.br/publico/extensions/AGROFIT/AGROFIT.html. The National Program for Control of Residues and Contaminants in Products of Vegetable Origin (PNCRC/Vegetal) contains analyses of residues such as arsenic, lead, cadmium, and mycotoxins from 2015 to 2020, and can be consulted in https://www.gov.br/agricultura/pt-br/assuntos/inspecao/produtos-vegetal/pncrcvegetal. and from PNCRC/Vegetal. Additionally, to incorporate information on water usage by crops (2013–2017), we accessed the National Water Resources Information System (SNIRH) on the National Water Agency of Brazil (ANA) platform, providing data on water usage, including needs, consumption, and deficits by crop, for the years 2013–2017, and available in https://www.snirh.gov.br/. Regarding to the details detected substances in food and specifies Maximum Residue Limits (MRL) in parts per million (ppm) or mg/kg, as well as Acceptable Daily Intake (ADI) in mg/kg/day, we use the data information according to National Health Surveillance Agency ANVISA/Mercosul Monographs available in: https://www.gov.br/anvisa/pt-br/setorregulado/regularizacao/agrotoxicos/monografias. A detailed description of all the agricultural platforms and databases used is presented below (Table 1).

**Nutrition data.** To calculate calories, macronutrients and micronutrients from the PAM database, the Food Composition Table (FCT) is produced by the Brazilian Institute of Geography and Statistics (IBGE) and provides detailed nutritional information per 100 grams of food, considering various preparation methods, available in: https://biblioteca.ibge.gov.br/index.php/biblioteca-catalog. Based on our goals, the selection of macro- and micronutrients was guided by the FAOSTAT Statistics Division's focus on energy and 17 key nutrients pertinent to food security within the 2030 Agenda for Sustainable Development [15]. Nutrients exclusive to animal sources (e.g., Vitamin B12) were excluded, while nutrients aligned with Brazilian consumption characteristics, such as manganese, copper, selenium, and folate, were added. [16–19].

Consumption information, by food, year, and geographical characteristics (including age groups and special conditions such as pregnancy), was extracted from the databases available in the Food and Nutrition Surveillance System (SISVAN), access on: https://sisaps.saude.gov.br/sisvan/relatoriopublico/index. Data on household food acquisition (per capita in kilograms), stratified by regions, states, and years of acquisition (2002, 2008, and 2018), were collected from the Family Budget Surveys (POF) databases prepared by the IBGE, in details on https://sidra.ibge.gov.br/. The description of nutrition databases used is presented below (Table 2).

## Results

Brazil has a vast amount of public information available through official government platforms. We selected eight platforms from which we collected and organized 114 datasets (Supplementary Material in S1 File), covering all five

**Table 1. Agricultural Data Providers, Sources, and Their Characteristics.**

| Data provider | Data source | Key Characteristics |
|---|---|---|
| Municipal Agricultural Production Information (PAM) | Municipal Agricultural Production | Crop records, Reference year (1974–2022), Harvested Area (ha), Quantity Produced (1,000 tons) |
| Ministry of Agriculture and Livestock (MAPA) | Phytosanitary Pesticides System (Agrofit) | The number of pesticides registered by year (1988–2023) and their functional classes. |
| | Phytosanitary Pesticides System (Agrofit) | Pesticides classified by the environmental and toxicity categories per crops |
| | National Program for Control of Residues and Contaminants in Products of Vegetable Origin – PNCRC/Vegetal | Residues per year (2015–2020), Type of analysis classification. |
| National Water Agency – Brazil (ANA) | National Water Resources Information System (SNIRH) | Information of the water uses (needs, consumption and deficits) by crops according to year (2013–2017) |
| National Health Surveillance Agency (ANVISA) | National Health Surveillance Agency (ANVISA) | Food, Substance detected, and the levels of residue control – Maximum Residue Limit (MRL), in parts per million (ppm) or mg.kg and Acceptable Daily Intake (ADI) in mg/kg/day |

This table lists the agricultural data providers utilized in the study, along with their respective data sources and key characteristics, including data coverage periods and specific metrics collected.

**Table 2. Nutrition Data Providers, Sources, and Their Characteristics.**

| Nutrition data provider | Data source | Characteristics |
|---|---|---|
| Brazilian Institute of Geography and Statistics (IBGE) | Food Composition Table (FCT) | Nutritional composition per 100 grams of food, considering various preparation methods. |
| Ministry of Health | Food and Nutrition Surveillance System (SISVAN) | Consumption data by food item, year (2015–2021), and demographic characteristics. |
| Brazilian Institute of Geography and Statistics (IBGE) | Family Budget Surveys (POF) | Household food acquisition data (kg per capita), by region, state, and year (2002, 2008, 2018). |

This table presents the nutrition data providers used in the study, detailing their data sources and key characteristics, such as data coverage periods and specific metrics collected.

regions (North, Northeast, Midwest, Southeast, South) and 27 federative units. The linkage process was structured into four phases, detailed below. In each phase, the data were compiled, filtered, harmonized, cleaned, and tracked by food name.

To reduce the risk of misinterpretation or misuse of the dataset, we designed a data structure explanation to be included both in the manuscript and the public repository. The dataset is organized in a wide format, with each row corresponding to a unique data point formed by horizontally aligned variables. During data harmonization, blank cells are maintained to indicate unavailable or inapplicable values, rather than being treated as zeros.

This structural approach follows international standards for food composition datasets, emphasizing metadata transparency, traceability of data origin, and the documentation of limitations, as recommended by FAO/INFOODS (2003), USDA Foundation Foods (2018), and the Brazilian TBCA (2019) [20–22]. The final outcome is a comprehensive Big Data (BD) resource, integrating information on 48 years of food production.

## Phase 1

Phase one consisted of linking the database of food produced (1,000 tons) and harvested area (ha) by year (1974–2022) from the Municipal Agricultural Production Information (PAM) (SM.Phase1 – Box 1) with the water information (needs, uses, and deficits) by crops according to year (2013–2017), regions, and districts from the National Water Agency – Brazil (ANA) (SM.Phase1 – Box 2). We also calculated the food nutrients produced using the Food Composition Table (FCT) from the Brazilian Institute of Geography and Statistics (IBGE) (SM.Phase1 – Box 3) of food produced.

Initially, the unstructured data from PAM and IBGE were extracted and organized into structured datasets, with names of variables and format harmonized for linkage. Non-edible agricultural (e.g., cotton, jute etc) products were then identified and filtered out from the PAM dataset, resulting in 45 food items classified as edible. The criteria for selecting FCT variables were aligned with the same objectives of our crop analysis, which excluded the food items and their nutrients from animal sources such as milk and eggs or Vitamin B12 or Vitamin D, or when the crudes vegetable sources weren't minimal, or the items weren't the focus of our goal (e.g., "SODIO DE ADIÇÃO"). Additionally, we opted to exclude analyses of aggregated foods, which include prepared foods with multiple components, industrialized products, and ultra-processed items. We also excluded items from the PAM dataset that were missing in the FCT, such as "CENTEIO", "GIRASSOL" and "MARMELO" excluded from the PAM dataset. To ensure consistency in scale, we calculated the nutrients from the FCT and converted the crop amounts from tons (as recorded in PAM) to grams.

Subsequently, the water file from SNIRH were integrated into the PAM and FCT datasets. The water data, were distributed across five datasets by year, detailing the water needs, uses, and deficits by crop. After extracting and filtering all relevant variables, the datasets were harmonized by food names, region, and state, resulting in the final linked dataset, referred to as Dataset 1. In total, seven datasets were utilized in this phase.

## Phase 2

In Phase 2, three datasets from the Phytosanitary Pesticides System (Agrofit) of the Ministry of Agriculture and Livestock (MAPA) were used for linkage (SM.Phase2 – Box 4). The first dataset contains information on pesticide registrations (including registration numbers and names) by year (1988–2023), along with their classifications based on pest control actions (e.g., weeds, pests, and diseases). The second dataset includes the registration numbers, pest control classes, active ingredients, and crop uses. The third dataset adds classifications for toxicity and associated environmental risks.

These three datasets were merged using pesticide names and codes. The crops were then filtered according to edible and non-edible agriculture products. The classification of toxicity and environmental risks were updated according to the most recent guidelines from the Brazilian National Health Surveillance Agency [23]. As a result, the Dataset 2 was integrated. These combined datasets enabled an analysis of the evolution of pesticide registrations in Brazil over time. Fig 2 presents two complementary visualizations: a circular (polar) bar chart illustrating the temporal distribution of pesticide registrations by toxicological classification, and a stacked bar chart showing the annual number of pesticide approvals categorized by environmental risk level.

## Phase 3

This phase involved organizing databases from the National Program for Control of Residues and Contaminants in Products of Vegetable Origin (PNCRC/Vegetal) under MAPA (SM.Phase3 – Box 5), and a database from ANVISA(SM.Phase3 – Box 6), resulting in a total of seven datasets (six from PNCRC/Vegetal and one from ANVISA). The PNCRC/Vegetal databases contain information on the adequacy of pesticide residues in food by year (2015–2020).

First, we harmonized the PNCRC/Vegetal databases by aligning the food names and creating a unified dataset that includes all food items and their residue adequacy information. Next, we added a new column to represent the corresponding years and then combined all the datasets. Finally, we harmonized the ANVISA dataset, which included food

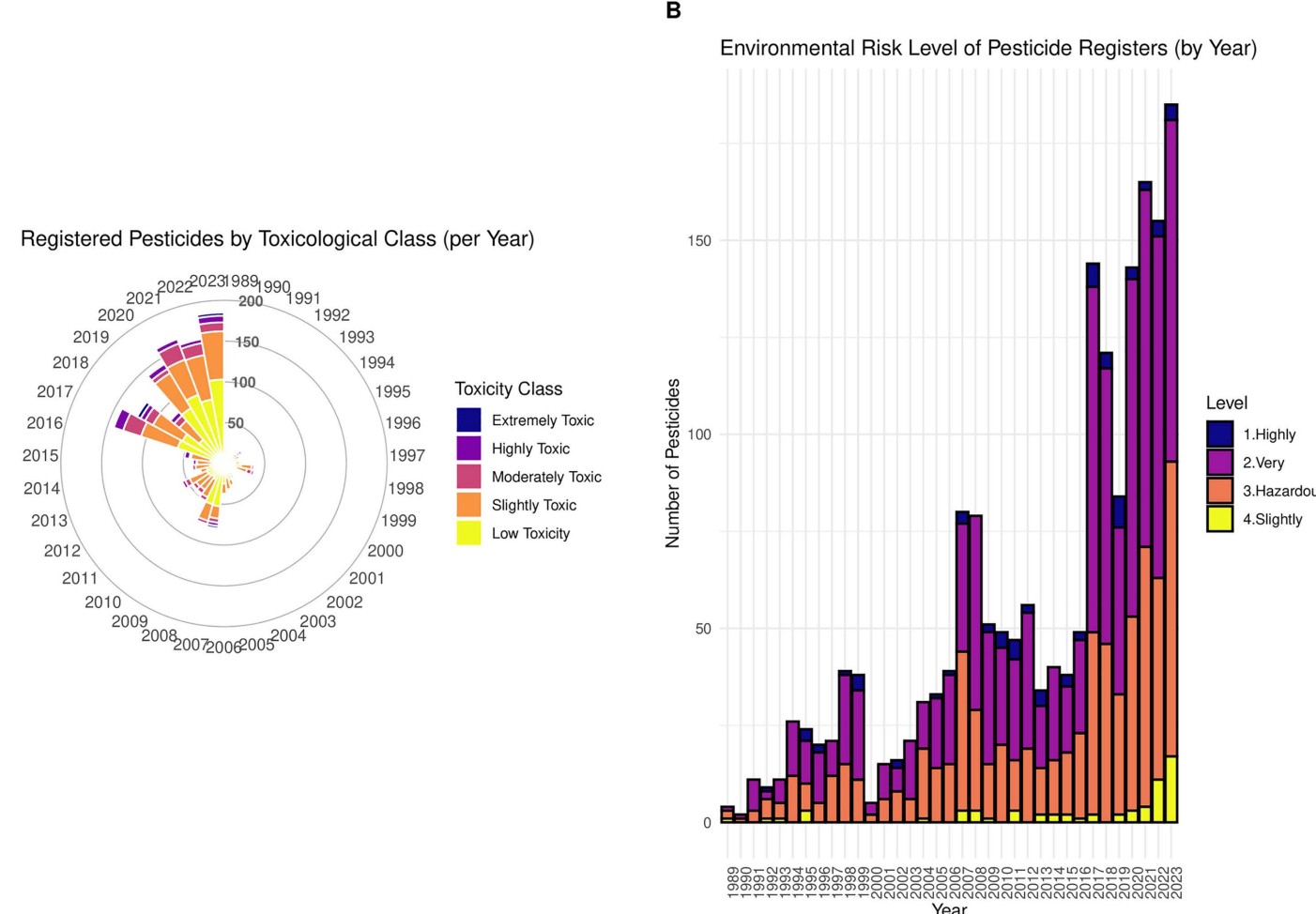

**Fig 2. Annual distribution of newly registered pesticides in Brazil by toxicological and environmental risk classification.** This figure combines two visualizations: (A) a circular bar chart displaying the number of pesticide registrations per year from 1988 to 2022, categorized by toxicological classification; and (B) a stacked bar chart representing the annual number of registrations by environmental risk level (1. Highly Hazardous, 2. Very Hazardous, 3. Hazardous, 4. Slightly Hazardous, and 5. Low Hazard). Together, these visualizations highlight temporal shifts in the intensity and composition of pesticide risk profiles in the country.

names, pesticides, and their Maximum Residue Limits (MRL), Acceptable Daily Intake (ADI). The reference year for the start of validity in determining the IDA and MRL limits was adjusted to the most updated reference from ANVISA. Afterwards, we integrated it with the unified PNCRC/Vegetal datasets, thus creating Dataset 3.

## Phase 4

Phase four focused on consumption information from the Food and Nutrition Surveillance System (SISVAN) (SM.Phase4 – Box 7-10) and the Family Budget Surveys (POF) (SM.Phase4 – Box 11). The SISVAN dataset, initially unstructured, was mined, structured, cleaned, and harmonized. We segmented the dataset by the number of respondents that answered to the "Yes or No" Dietary Questionnaires called "Food Consumption Markers" about their consumption of household availability. The questionnaire focuses on specific food groups: beans, fresh fruits (excluding fruit juices), and vegetables/legumes (excluding potatoes, cassava, yams, and other starchy roots).

The term "food groups" in this phase refers to these dietary categories based on nutritional groupings. This differs from the term "vegetable crops", which is an agricultural classification encompassing crops such as corn, oats, rice, wheat, beans, peas, peanuts, soy, cassava, potatoes, coffee, sugarcane, palm heart, scarlet eggplant, tomato, açaí, apple, avocado, banana, cashew, coconut, fig, grape, guava, lime, mango, melon, papaya, passion fruit, peach, pear, persimmon, pineapple, and watermelon.

The databases were also segmented by regions, districts, age groups, and special conditions (e.g., teens, adults, elderly, and pregnant individuals) over six years (2015–2022), resulting in 24 datasets for each food group (beans, fruits, and vegetables), totalling 96 datasets, which were consolidated into one comprehensive database.

Afterwards, we worked with the unstructured data from the Family Budget Surveys (POF) for the years 2002, 2008, and 2018. To harmonize POF dataset with all other datasets, we selected only total intake across multiple regions. The POF dataset was transformed into a structured format, with food names aligned and special characters cleaned to facilitate integration into Dataset 4. To illustrate the relationship between household-level food acquisition and individual dietary intake, we developed two parallel Sankey diagrams (Fig 3).

## PHFood database

The processes of the final PHFood BD encompass two key steps. First, we aligned and merged the pesticide and food names between Dataset 2 and Dataset 3. Following this, we harmonized and merged Dataset 1 with Dataset 4 by food names or food groups and year. All the information contained in the original datasets was preserved, and we decided not to input missing values for years or food items. In this phase, foods were grouped based on their presence across the

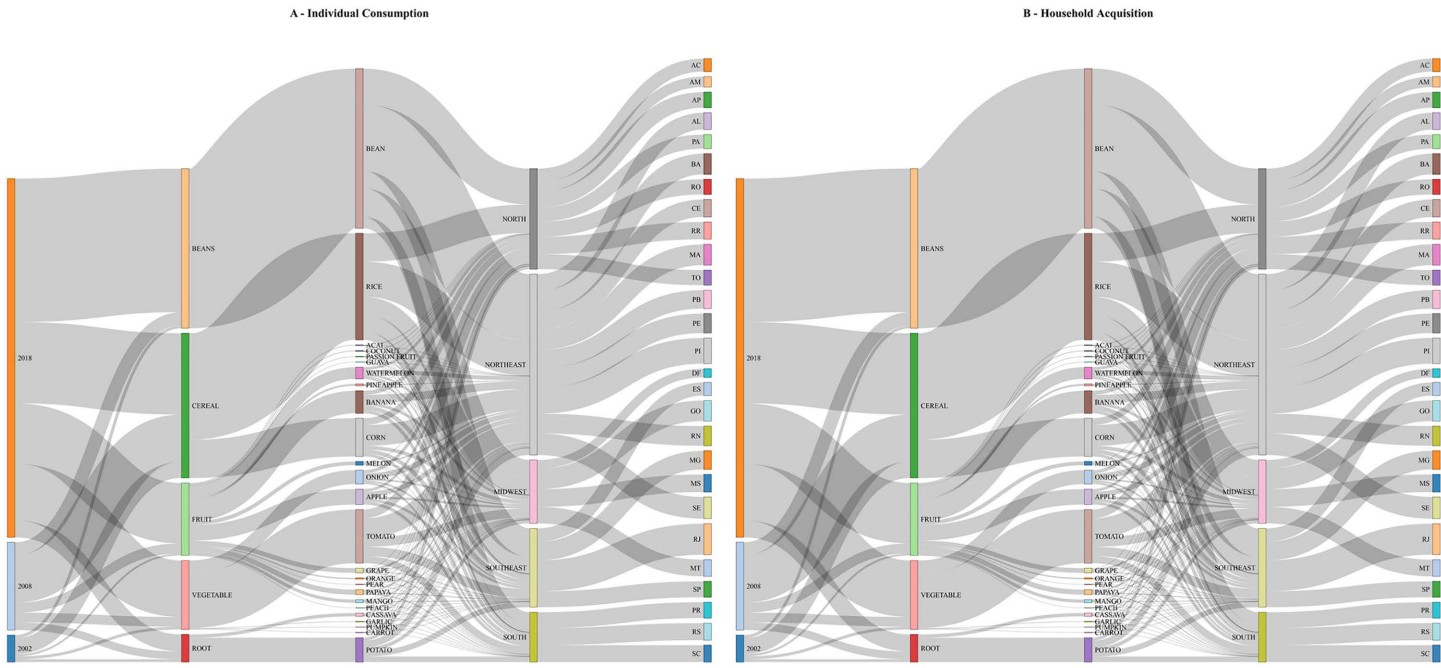

**Fig 3. Parallel Visualizations of Brazilian Food Acquisition and Consumption.** The first visualization (A) displays the distribution of food group consumption across the population, using data from Brazil's national food and nutrition surveillance system (SISVAN). The second visualization (B) represents the 15 most acquired foods by Brazilian families, based on aggregated data from the Household Budget Survey (POF). These diagrams were harmonized and presented side by side to offer a comparative perspective on food flow structures, from purchase to consumption, across different food categories and regions. Link: http://rpubs.com/MariaMiele/1313974.

unified datasets from all four phases; in line with the objectives of creating this database, any food item that appeared in only one phase, without presence in the others, was excluded from the final merge. Finally, these combined datasets were integrated into the PHFood historical series. This comprehensive BD spans 48 years of food data across five regions and 27 federative units in Brazil.

As a result of the integrated datasets, an interactive visualization was developed to illustrate temporal trends in total plant-based food production, harvested area, energy supply, and pesticide registrations in Brazil from 1974 to 2022 (Fig 4).

This tool highlights the applicability of PHFood Brazil in exploring long-term patterns and facilitating public access to open data.

## Discussion

The FAIR concept [24] is fundamental to ensuring transparency in science, and this study has detailed all the steps and sources used to create a Big Data on Brazil's food history. To our knowledge, no previous research has integrated such an extensive array of datasets on food production and consumption.

Data is generated and collected on a massive scale every day, leading to a vast number of datasets stored in data warehouses. However, the agricultural community faces a growing challenge in effectively compiling and analysing this large volume of data [25]. The procedures employed in this study represent an innovative approach, linking nearly five decades of information on plantation, food and nutrients from 114 open-source datasets across eight platforms. The goal is to learn from Brazil's food production and consumption patterns, given its status as one of the world's largest agricultural producers. This knowledge will help address planetary boundaries and the challenges of providing healthy food while avoiding hunger.

Historical data plays a vital role in identifying trends and patterns in agricultural practices, food production, and consumption, driving strategic thinking towards a sustainable, resilient, and inclusive future [26]. By examining extensive data, we can observe changes in food production and project potential hunger scenarios in the near future [27]. The complexity of Big Data, which involves enormous volumes of structured and unstructured data from heterogeneous sources, makes data harmonization crucial [28]. Our approach will provide a clear and comprehensive view of Brazil's food history, with the resulting Big Data housed in an open repository, offering valuable insights and benefits to various stakeholders.

Data is crucial for guiding public policy and informing investment decisions aligned with the Sustainable Development Goals (SDGs). However, challenges such as inconsistent investment in data, limited technical skills, and difficulties accessing new data sources hinder progress in the agri-food system [29]. The use of data science techniques to harmonize and analyze these datasets enhances the granularity and accessibility of information, facilitating a deeper understanding of the complex interactions between food production, environmental sustainability, and future pathways [11]. In this study, by linking data to specific foods, we will be able to assess variations in the consumption of certain foods in response to changes in crop production.

This study has limitations that should be considered. One limitation is the reliance on the Brazilian Food Composition Table (FCT) for nutritional data, which assumes that the nutritional content of foods has remained constant from 1980 to 2022. This approach may not account for changes in agricultural practices, soil health, and crop varieties, all of which can affect nutrient levels. Future research should incorporate direct analyses of crops alongside the FCT data to provide a more accurate assessment of nutrient content over time.

The key strength of this study lies in its historical depth, covering food production pathways from 1974 to 2022 across regions and states. This extensive time frame allows for the identification of long-term pesticide uses and residues in food, which is crucial for developing strategies to ensure sustainable food systems and food security. Additionally, the detailed water usage data offers valuable insights into the water footprint of different crops, aiding in the development of water-efficient agricultural practices.

## Total Plant-Based Food Production, Land Use, Pesticide Registrations, and Energy Supply in Brazil (1974–2022)

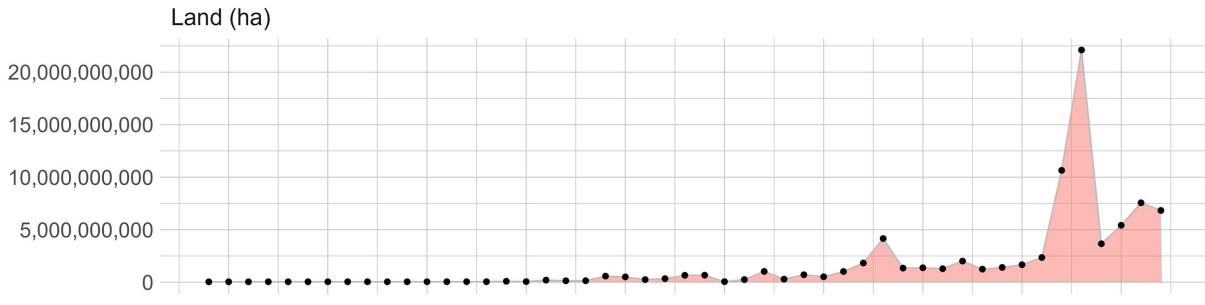

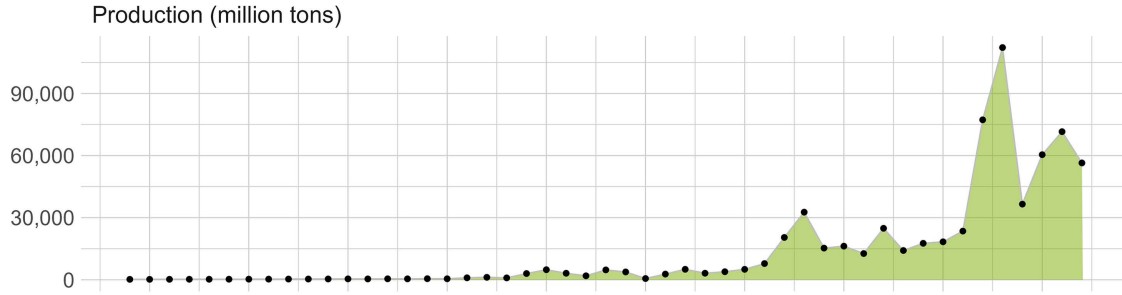

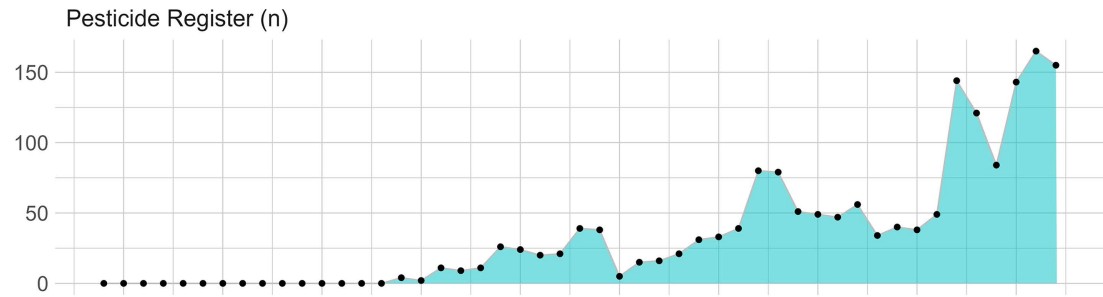

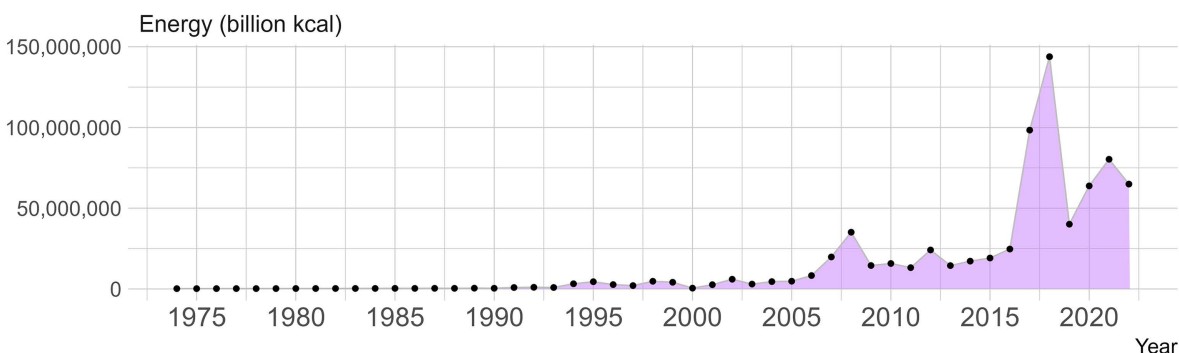

**Fig 4. Total Plant-Based Food Production, Land Use, Pesticide Registrations, and Energy Supply in Brazil (1974–2022). link: https://rpubs.com/MariaMiele/1323568.**

The long-term relevance of the PHFood Brazil database depends on its capacity to evolve alongside national data systems. As the source platforms (e.g., IBGE, ANVISA, Ministry of Agriculture) release new data, future updates to the database are planned to reflect these changes. This ongoing maintenance will allow the integration of new variables, improved harmonization procedures, and broader analytical applications, ensuring that the database remains a useful tool for research, monitoring, and policy development.

In conclusion, this work outlines the processes for building a valuable tool called PHFood Brazil, a historical series of data on food production and consumption in Brazil, viewed through the lens of agricultural and food processes. By understanding the past and present dynamics of food production and consumption, we can better navigate the challenges of ensuring food security and environmental sustainability for future generations.

## Supporting information

**S1 File. Supplementary Material.** Includes the datasets and metadata used in the PHFood Brazil integration process, structured in four phases: Phase 1 – Datasets of food production and land use from the Municipal Agricultural Production Information (PAM) by year; datasets from the National Water Agency (ANA); and the Brazilian Food Composition Table (IBGE). Phase 2 – Datasets from the Ministry of Agriculture and Livestock (MAPA), including the Phytosanitary Pesticides System (Agrofit). Phase 3 – Datasets from the National Plan to Control Residues and Contaminants (PNCRC/Vegetal) and the National Health Surveillance Agency (ANVISA). Phase 4 – Files from the Food and Nutrition Surveillance System (SISVAN) for adults, special conditions, elderlies, and teenagers, and the Family Budget Surveys (POF) for food acquisition data. (DOCX)

## Acknowledgments

The author is grateful for the support provided by the São Paulo Center for the Study of Energy Transition (CPTEn) for its institutional support and the Applied Research Center in Artificial Intelligence BI0S – Brazilian Institute of Data Science.

## Author contributions

**Conceptualization:** Maria Julia Miele, Barbara Teruel.

**Data curation:** Maria Julia Miele.

**Formal analysis:** Maria Julia Miele.

**Methodology:** Maria Julia Miele, Jacqueline Tereza da Silva, Barbara Teruel.

**Project administration:** Maria Julia Miele, Barbara Teruel.

**Supervision:** Barbara Teruel.

**Validation:** Maria Julia Miele, Jacqueline Tereza da Silva, Renato Teixeira Souza, José Guilherme Cecatti, Barbara Teruel.

**Visualization:** Jacqueline Tereza da Silva.

**Writing – original draft:** Maria Julia Miele.

**Writing – review & editing:** Maria Julia Miele, Jacqueline Tereza da Silva, Renato Teixeira Souza, José Guilherme Cecatti, Barbara Teruel.

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
