## [Decision Letter · Decision Letter 0]

19 May 2025

Dear Dr. Miele,

Thank you for submitting your manuscript to PLOS ONE. After careful consideration, we feel that it has merit but does not fully meet PLOS ONE’s publication criteria as it currently stands. Therefore, we invite you to submit a revised version of the manuscript that addresses the points raised during the review process.

We look forward to receiving your revised manuscript.

Kind regards,

António Raposo

Academic Editor

PLOS ONE

Journal Requirements:

Reviewers' comments:

Reviewer's Responses to Questions

**Comments to the Author**

1. Is the manuscript technically sound, and do the data support the conclusions?

Reviewer #1: Yes

Reviewer #2: Yes

Reviewer #3: Yes

2. Has the statistical analysis been performed appropriately and rigorously?

Reviewer #1: N/A

Reviewer #2: No

Reviewer #3: N/A

3. Have the authors made all data underlying the findings in their manuscript fully available?

Reviewer #1: Yes

Reviewer #2: Yes

Reviewer #3: Yes

4. Is the manuscript presented in an intelligible fashion and written in standard English?

Reviewer #1: Yes

Reviewer #2: No

Reviewer #3: Yes

Reviewer #1: The study presents the development of a robust database (named PHFood database) that integrates spatiotemporal information on food production and consumption in Brazil. This database enables the identification of the types of foods produced, pesticide and water use, and their associated nutritional values across space and time. Such comprehensive information provides a valuable foundation for advancing research on biodiversity conservation, food security, and sustainable development.

The data is well-organized and highlights the importance of integrating different public databases to achieve a more comprehensive understanding of food systems in the country. The description of the methods is well-structured, and the results are presented in an organized manner.

The authors should provide at least one practical example illustrating the usefulness of this data, whether through graphs or maps. The knowledge compiled by the authors about Brazil can demonstrate the importance of keeping this repository updated and, consequently, ensuring the continuous supply of data sources and investment in the responsible institutions.

In the description of the dataset available in the Mendeley repository (DOI: 10.17632/mt4mj23j73.9), it is evident that the authors analyze the dataset based on a hypothesis. Part of this text should be incorporated into the introduction and/or discussion of the main article.

If the authors take, for example, a specific food item from the dataset and demonstrate how integrating this data into Big Data environments can contribute to understanding patterns of pesticide use over time, agricultural productivity, and other relevant issues, the study will expand its reach across different scientific fields and attract new users to the dataset.

The authors should also briefly discuss, in their final discussion, how this database can be maintained and updated in the future.

Reviewer #2: The article is relevant for its proposal to integrate public data and promote transparency in data science applied to food and health in Brazil. However, from a methodological point of view, it has substantial weaknesses that compromise its validity as scientific research. 1) Although the article has a broad introduction, there is no explicit formulation of hypotheses, research questions or specific objectives beyond the construction of the PHFood database. For a scientific publication, it is essential to clearly explain the objectives and justify how the methodology responds to them.

2. The calculation of nutrients was based on fixed values from the Food Composition Table (FCT), ignoring possible temporal variations (1974 to 2022) in nutritional content due to:

Changes in cultivars;

Climate change;

Soil depletion;

Changes in cultivation and harvesting methods. I suggest that the authors justify this.

3. What is the justification for integrating the agricultural production (PAM), consumption (SISVAN, POF), pesticide use (Agrofit) and water quality (SNIRH) databases? It is important to justify the integration of these databases, given that the databases have different spatial and temporal scales (e.g. annual production vs. consumption per three-year period); the link “by food name” is fragile and can generate inaccurate or invalid correspondences; some foods were excluded because they did not have an exact correspondence between the databases, which can introduce self-reporting bias.

4 The study excludes foods of animal origin, justifying it only by their absence in some bases. However, these foods are essential for the analysis of food and nutritional security. I recommend their inclusion.

5. Although the PHFood database is advertised as a tool for public policy, no statistical analysis has been carried out using the integrated database to demonstrate its applicability. The lack of practical examples of how to use the data reduces the article's relevance as a scientific study and brings it closer to a technical report.

Reviewer #3: Overall, a very interesting read. Congratulations to the authors! This work provides a very useful temporal and spatial view of Brazilian agri-food systems and the database has clear relevance for public health, agricultural policy, and planetary health research.

Some small suggestions follow:

1. Regarding relevance and contribution, authors could clarify that the paper’s focus is methodological (i.e., on data integration) rather than on generating new empirical findings;

2. Why not add a brief proof-of-concept analysis, like a trend line of pesticide residues over time for one major crop, or nutrient supply variation across regions? This would illustrate the potential of the dataset; suggest typical research questions that could be answered with PHFood Brazil;

3. What does this platform enables that was previously impossible? Regional comparisons, predictive modeling, time-series forecasting?

4. Please, comment on the fact that the work assumes nutrient composition is stable over time… This may be problematic over a 48-year window;

5. It would be useful to include the dataset’s versioning and update plan. How will future data be integrated or revised?

6. Perhaps the authors could briefly reflect on the risks of public dataset misuse or misinterpretation.

**Do you want your identity to be public for this peer review?** For information about this choice, including consent withdrawal, please see our Privacy Policy

Reviewer #1: No

Reviewer #2: **Yes: ** Nathalia Sernizon Guimarães

Reviewer #3: No

---

## [Author Response · Author response to Decision Letter 1]

10 Jul 2025

RESPONSE TO REVIEWERS

Reviewer #1

1. The authors should provide at least one practical example illustrating the usefulness of this data, whether through graphs or maps. The knowledge compiled by the authors about Brazil can demonstrate the importance of keeping this repository updated and, consequently, ensuring the continuous supply of data sources and investment in the responsible institutions.

We are thankful for this suggestion and, in response, have added illustrative figures to demonstrate the functionality and usability of the PHFood Brazil database across time and space. These visualizations highlight the analytical potential of the platform and illustrate how the integrated data can reveal long-term trends relevant to planning for sustainable agricultural production and nutritional security.

• Figure 2 was included in the Results section, Phase 2 (in lines 256 - 271): presents the annual number of pesticide registrations in Brazil, grouped by toxicological and environmental risk categories, highlighting trends in potential exposure risks over time.

• Figure 3 was included in the Results section, Phase 4 (in lines 314 - 325). compares data from the POF and SISVAN systems to showcase food consumption and acquisition shifts across the population. This figure illustrates how changes in food production may align with variations in dietary patterns over time. Link: http://rpubs.com/MariaMiele/1313974

• Figure 4 was included in the Results section, Final Phase (line 339 - 346) - PHFood database presents a time-series visualization of total plant-based food production, harvest area, pesticide registrations, and caloric availability (kcal) in Brazil from 1974 to 2022. Link: https://rpubs.com/MariaMiele/1323529

•

We also reinforce our commitment to keeping the repository updated and progressively expanding its coverage. The versioning and update plan has been detailed both in the public repository and in the Discussion section of the manuscript.

2. In the description of the dataset available in the Mendeley repository (DOI: 10.17632/mt4mj23j73.9), it is evident that the authors analyze the dataset based on a hypothesis. Part of this text should be incorporated into the introduction and/or discussion of the main article.

We thank the reviewer for this observation. In response, we incorporated part of the rationale originally included in the dataset repository description into the Introduction (in lines 80 – 88) and Material and Methods (in lines 94, 95, 98 – 100). This addition clarifies that the PHFood Brazil database was developed under the hypothesis that harmonizing food production, consumption, and environmental data over time would enable the identification of relevant patterns and relationships across food systems. This clarification strengthens the methodological focus of the manuscript and supports future analytical applications of the dataset.

3. The authors should also briefly discuss, in their final discussion, how this database can be maintained and updated in the future.

We thank the reviewer for this important observation. In response, we added a brief paragraph in the final Discussion section addressing how the PHFood Brazil database can be maintained and updated in the future (in line 395 – 401). The database was designed as an extensible platform, and future updates are planned to follow a structured versioning process aligned with the release schedules of the original data providers. This approach ensures the continued relevance, transparency, and reproducibility of the dataset.

Reviewer #2

1. Although the article has a broad introduction, there is no explicit formulation of hypotheses, research questions or specific objectives beyond the construction of the PHFood database. For a scientific publication, it is essential to clearly explain the objectives and justify how the methodology responds to them.

We appreciated the reviewer's comment and agreed that the original version of the manuscript did not formulate the specific objectives of the study beyond the construction of the PHFood Brazil database. In response, we revised both the Introduction and the Methods sections to explicitly state that the main objective of the study is methodological (in lines 80-88, 94, 95): to describe, justify, and document the process of integrating and harmonizing open-access datasets related to food production, consumption, and environmental indicators in Brazil.

2. The calculation of nutrients was based on fixed values from the Food Composition Table (FCT), ignoring possible temporal variations (1974 to 2022) in nutritional content due to: Changes in cultivars;Climate change;Soil depletion; Changes in cultivation and harvesting methods. I suggest that the authors justify this.

We acknowledge that assuming stability in nutrient composition over a 48-year period represents a limitation, as noted in lines 381–387 of the manuscript. This approach does not reflect potential variations arising from factors such as changes in cultivars, climate variability, soil nutrient depletion, and evolving cultivation and harvesting practices.

However, longitudinal data on food nutrient composition spanning multiple decades is not currently available in Brazil or globally. In the absence of such data, we followed the standard procedure adopted by international institutions established guidance from leading institutions such as the Food and Agriculture Organization (FAO) and the United States Department of Agriculture (USDA). These organizations, including FAO/INFOODS and the USDA’s Foundation Foods system, frequently rely on representative nutrient values in cases where longitudinal analytical data are unavailable.

3. What is the justification for integrating the agricultural production (PAM), consumption (SISVAN, POF), pesticide use (Agrofit) and water quality (SNIRH) databases? It is important to justify the integration of these databases, given that the databases have different spatial and temporal scales (e.g. annual production vs. consumption per three-year period); the link “by food name” is fragile and can generate inaccurate or invalid correspondences; some foods were excluded because they did not have an exact correspondence between the databases, which can introduce self-reporting bias.

We appreciate the observation. The integration of agricultural production (PAM), food consumption (SISVAN, POF), pesticide registration (Agrofit), and water quality and availability (SNIRH) data were guided by the aim of providing a more systemic and interdisciplinary view of Brazilian food systems. Although the original datasets differ in temporal and spatial resolution, they each contribute essential dimensions - production, intake, environmental impact, and safety; that are rarely analyzed together but are highly interdependent in food system assessments.

We acknowledge the methodological challenges of linking datasets with different structures. To mitigate inconsistencies, all integration steps were carried out using a carefully designed ETL process, and all linkages were made only when robust matches were possible. The correspondence “by food name” was manually verified and standardized across datasets to reduce ambiguity, and items that lacked valid correspondence across datasets were excluded to preserve internal consistency. These exclusions are documented and were applied uniformly across all sources, minimizing the introduction of bias.

Additionally, we added these limitations acknowledged in the Methods and Results (in lines 98-100 and 201 – 210) sections of the manuscript. We also included a description of these filtering decisions in the public repository and the accompanying codebook to support transparency and reproducibility.

4. The study excludes foods of animal origin, justifying it only by their absence in some bases. However, these foods are essential for the analysis of food and nutritional security. I recommend their inclusion. In the next version. This project is a continuum and the dataset’s versioning and update plan. How will future data be integrated or revised?

We appreciate the reviewer’s comment and agree that foods of animal origin are essential for comprehensive food and nutritional security assessments. In this initial version of the PHFood Brazil database, the focus was placed on plant-based foods due to the relevance of better understanding the agricultural producers' practices.

However, we acknowledge the relevance of this food group, and, in line with our versioning and update plan, future releases of the database are expected to incorporate additional sources and expand the scope to include animal-based foods, contingent upon data availability and quality. These updates will follow the same transparent, traceable ETL process described in this version. All changes and additions will be documented in the repository with clear version identifiers and updated codebooks to ensure reproducibility and user confidence.

5. Although the PHFood database is advertised as a tool for public policy, no statistical analysis has been carried out using the integrated database to demonstrate its applicability. The lack of practical examples of how to use the data reduces the article's relevance as a scientific study and brings it closer to a technical report.

We appreciate the reviewer’s comment and recognize the importance of demonstrating the applicability of the PHFood Brazil database. This article focuses on presenting the methodological framework for integrating and harmonizing public open-access datasets related to food production, consumption, and environmental indicators in Brazil. The primary aim was not to perform statistical modeling, but to document the integration process and provide a robust data infrastructure.

However, to offer a perspective on the potential applications of the database, we included a series of illustrative visualizations. Figures 2, 3, and 4 (in lines 263, 322 and 341) demonstrate how the integrated data can be used to explore time trends, regional disparities, and cross-sectoral relationships among variables such as food acquisition, pesticide registration, and agricultural production. These examples highlight the value of the PHFood Brazil database as a foundational tool for informing public policies, monitoring food systems, and supporting sustainability analyses.

Reviewer #3:

1. Regarding relevance and contribution, authors could clarify that the paper’s focus is methodological (i.e., on data integration) rather than on generating new empirical findings

We appreciate the suggestion, as the main objective of our work is methodological. Accordingly, the manuscript was revised to clarify that its primary focus is to describe and detail the process of compiling open-access public datasets from diverse national sources, rather than presenting new empirical findings. This clarification was added to both the Introduction and the Methods sections (in lines 80 – 83, 94 – 95, 98 – 100).

2. Why not add a brief proof-of-concept analysis, like a trend line of pesticide residues over time for one major crop, or nutrient supply variation across regions? This would illustrate the potential of the dataset; suggest typical research questions that could be answered with PHFood Brazil;

We appreciate the reviewer’s suggestion to include a brief proof-of-concept analysis to demonstrate the dataset’s potential. In response, we added illustrative visualizations to the manuscript that highlight possible lines of investigation enabled by PHFood Brazil.

Figure 2 (line 263) presents a multi-variable visualization linking total plant-based food production, land use, pesticide registrations, and energy supply from 1974 to 2022.

Figure 3 (in line 317) shows food acquisition and consumption patterns across the population using POF and SISVAN data, enabling regional and temporal comparisons.

Figure 4 (in line 343) demonstrates the integration of key variables across space and time, offering a model of how the database can be used to examine trends, disparities, and potential associations.

Together, these visualizations serve as practical demonstrations of the platform’s capacity to support trend analysis, comparative research, and the formulation of policy-relevant questions. Future work may build on this foundation to conduct crop-specific or nutrient-specific time-series analyses, as suggested.

3. What does this platform enables that was previously impossible? Regional comparisons, predictive modeling, time-series forecasting?

We appreciate the reviewer’s question. The PHFood Brazil platform enables several analytical capabilities that were previously not feasible due to data fragmentation, format incompatibilities, or the lack of harmonization among public datasets.

By integrating and aligning variables such as food production, water footprint, pesticide use, nutrient composition, and food acquisition over a 48-year period, the platform allows for:

• Temporal evaluations of how agricultural practices, food availability and costs, and dietary patterns have evolved over time;

• Regional comparisons across Brazil’s five regions and 27 federative units, making it possible to analyze spatial disparities and heterogeneities in food systems;

• Cross-variable integration, enabling the examination of relationships between production systems (e.g., land use, food production, water requirements and usage over the years) and public health dimensions (e.g., food group consumption and pesticide registrations categorized by toxicological and environmental risk by crop);

A foundation for advanced modeling, including time-series forecasting, trend detection, and machine learning applications, which were previously limited by inconsistent or siloed datasets. To illustrate these new capabilities, we added new figures to the manuscript, which presents integrated data across time and space, exemplifying the core functionality and usability of the PHFood Brazil database. These illustrations demonstrate how the platform supports comparative, longitudinal, and systems-based assessments that were not possible prior to this harmonization effort.

4. Please, comment on the fact that the work assumes nutrient composition is stable over time… This may be problematic over a 48-year window;

We acknowledge that assuming stability in nutrient composition over a 48-year period is a limitation, as noted in lines 377–383 of the manuscript. This assumption does not account for potential changes over time due to factors such as changes in cultivars, climate variability, soil nutrient depletion, and evolving cultivation and harvesting practices. However, to address this concern, we have aligned our approach with established guidance from leading institutions such as the Food and Agriculture Organization (FAO) and the United States Department of Agriculture (USDA). These organizations, including FAO/INFOODS and the USDA’s Foundation Foods system, frequently rely on representative nutrient values in cases where longitudinal analytical data are unavailable (in lines 207 – 210).

5. It would be useful to include the dataset’s versioning and update plan. How will future data be integrated or revised?

We included in the Discussion section a versioning and update plan for the PHFood Brazil dataset (in lines 395 – 401). This framework enables the incorporation of revised nutrient composition values in future updates as new analytical data become available, ensuring the continued accuracy and relevance of the database.

6. Perhaps the authors could briefly reflect on the risks of public dataset misuse or misinterpretation

Thank you for this observation. We agree with the reviewer to further support transparency and appropriate data use; we incorporated a reflection on the risks of dataset misinterpretation directly into the Results section of the manuscript (in lines 201 – 206). Specifically, we included a description of the data structure, its limitations, and guidance for appropriate use. This includes clarifying the wide-format organization, the treatment of missing data (e.g., blank cells not representing zeros), and the importance of considering only aligned rows as valid data points.

---

## [Decision Letter · Decision Letter 1]

16 Jul 2025

Integrating national open databases for a comprehensive view on food systems, environment sustainability and health in Brazil

PONE-D-25-07539R1

Dear Dr. Miele,

We’re pleased to inform you that your manuscript has been judged scientifically suitable for publication and will be formally accepted for publication once it meets all outstanding technical requirements.

Kind regards,

António Raposo

Academic Editor

PLOS ONE

Additional Editor Comments (optional):

Reviewers' comments:

Reviewer's Responses to Questions

**Comments to the Author**

Reviewer #2: All comments have been addressed

Reviewer #3: All comments have been addressed

2. Is the manuscript technically sound, and do the data support the conclusions?

Reviewer #2: Yes

Reviewer #3: Yes

3. Has the statistical analysis been performed appropriately and rigorously?

Reviewer #2: Yes

Reviewer #3: Yes

4. Have the authors made all data underlying the findings in their manuscript fully available?

Reviewer #2: Yes

Reviewer #3: Yes

5. Is the manuscript presented in an intelligible fashion and written in standard English?

Reviewer #2: Yes

Reviewer #3: Yes

Reviewer #2: I congratulate the authors on their current and well-conducted work. All responses were satisfactory. I believe that the article can be published.

Reviewer #3: (No Response)

**Do you want your identity to be public for this peer review?** For information about this choice, including consent withdrawal, please see our Privacy Policy

Reviewer #2: **Yes: ** Nathalia Sernizon Guimarães

Reviewer #3: No

---

## [Editor Report · Acceptance letter]

PONE-D-25-07539R1

PLOS ONE

Dear Dr. Miele,

I'm pleased to inform you that your manuscript has been deemed suitable for publication in PLOS ONE. Congratulations! Your manuscript is now being handed over to our production team.

Kind regards,

on behalf of

Dr. António Raposo

Academic Editor

PLOS ONE